# Shotgun Metagenomic Sequencing Revealed the Prebiotic Potential of a Fruit Juice Drink with Fermentable Fibres in Healthy Humans

**DOI:** 10.3390/foods12132480

**Published:** 2023-06-25

**Authors:** Adri Bester, Marcus O’Brien, Paul D. Cotter, Sarita Dam, Claudia Civai

**Affiliations:** 1London Agri Food Innovation Clinic (LAFIC), School of Applied Sciences, London South Bank University, London SE1 0AA, UK; civaic@lsbu.ac.uk; 2SeqBiome, P61C996 Cork, Ireland; marcusobrien@seqbiome.com (M.O.); paulcotter@seqbiome.com (P.D.C.); saritadam@seqbiome.com (S.D.)

**Keywords:** shotgun metagenomics, gut microbiota, prebiotics, *Bifidobacterium adolescentis*, *Lachnospiraceae*, arginine, fibre

## Abstract

Fibre-based dietary interventions are at the forefront of gut microbiome modulation research, with a wealth of 16S rRNA information to demonstrate the prebiotic effects of isolated fibres. However, there is a distinct lack of data relating to the effect of a combination of soluble and insoluble fibres in a convenient-to-consume fruit juice food matrix on gut microbiota structure, diversity, and function. Here, we aimed to determine the impact of the MOJU Prebiotic Shot, an apple, lemon, ginger, and raspberry fruit juice drink blend containing chicory inulin, baobab, golden kiwi, and green banana powders, on gut microbiota structure and function. Healthy adults (*n* = 20) were included in a randomised, double-blind, placebo-controlled, cross-over study, receiving 60 mL MOJU Prebiotic Shot or placebo (without the fibre mix) for 3 weeks with a 3-week washout period between interventions. Shotgun metagenomics revealed significant between-group differences in alpha and beta diversity. In addition, the relative abundance of the phyla *Actinobacteria* and *Desulfobacteria* was significantly increased as a result of the prebiotic intervention. Nine species were observed to be differentially abundant (uncorrected *p*-value of <0.05) as a result of the prebiotic treatment. Of these, *Bifidobacterium adolescentis* and CAG-81 sp900066785 (*Lachnospiraceae*) were present at increased abundance relative to baseline. Additionally, KEGG analysis showed an increased abundance in pathways associated with arginine biosynthesis and phenylacetate degradation during the prebiotic treatment. Our results show the effects of the daily consumption of 60 mL MOJU Prebiotic Shot for 3 weeks and provide insight into the functional potential of *B. adolescentis.*

## 1. Introduction

Research involving interventions designed to modulate gut microbiota with the aim to improve host health has contributed to accruing evidence of a bidirectional gut microbiota–host axis. Driven by technological advances and reductions in the costs associated with analyses, important advances have been made in identifying microbial communities and their influence on host health. We are now able to ask “who is there” and “what can they do” with the aid of shotgun metagenomics that provide more detailed microbiome insights [1,2].

The collection of microorganisms inhabiting the mammalian gastrointestinal tract is now considered to have a central position in health and disease, with the ability to affect organs far beyond the abdomen [3,4,5]. This complex community that includes bacteria, archaea, viruses, fungi, and protozoa is fundamental to maintaining host physiological homeostasis [6], including programming host circadian rhythms [7,8,9]. Many of these gut microbes participate in several beneficial-to-host functions, including essential metabolic outputs such as short-chain fatty acids (SCFAs) [10] and neurotransmitters [11].

Diet is widely agreed to be a key modulator of the gut microbiota. This is particularly true for dietary fibre, broadly defined as those carbohydrate polymers and oligomers (plus lignin) that escape digestion in the small intestine, passing into the large bowel where they are partially (insoluble dietary fibre) or more completely (soluble dietary fibre) fermented and metabolised by the gut microbiota [12]. Reaching the large intestine, where the highest concentration of gut microbiota is located at an abundance of around 10^11^ bacterial cells per mL of content [13], dietary fibres are fermented into products that can provide a variety of localised and systemic health benefits to the host [14]. Such is the interest in the impact of dietary fibre on human health, ongoing work supported by the Technical Committee on Dietary Carbohydrates of the North American Branch of the International Life Sciences Institute (ILSI North America) and Tufts University has resulted in a publicly available research database linking fibres to a variety of health outcomes [15]. Recognised health benefits associated with dietary fibre consumption, and which have been extensively reviewed [16,17], include improved gastrointestinal function [18], moderation of circulating blood lipids [19], and improvement of post-prandial serum glucose and insulin responses [20,21,22]. Furthermore, chronic low intake of dietary fibre has been associated with many deleterious health consequences including increased risk of colorectal cancer [23,24] diverticular disease [25], cardiovascular disease [26], obesity [22], metabolic syndrome [14], pre-diabetes, and type-2 diabetes [23].

The definition of dietary fibres differs between countries, as detailed elsewhere [14,27,28]. Fibre solubility (dissolution capability in water), viscosity (gelling capability in water), and fermentability (degree of metabolising capability by gut microbiota) are among the factors that determine the effect of fibre on the gut microbiota and host [29]. Despite decades of lifestyle and dietary advice and education on the importance of adequate fibre in the diet, our diets continue to lack sufficient fibre [14,27,30], even in Mediterranean countries [31]. Latest data for the UK and US for adults 19 years and over (gender combined) suggest consumption of 19.7 and 16.6 g/day, respectively [32,33]. Dietary fibre intakes around the world are described in detail elsewhere [14,27,30,34]. Ultimately, intake levels are low considering the fact that the European Food Safety Authority (EFSA) and the Food and Agriculture Organization and World Health Organization (FAO/WHO) recommend the consumption of 25 g of dietary fibre per day for adults while the recommended intake in the UK is 30 g per day [35]. Recently, it has been proposed that consumption of over 50 g fibre per day is needed to achieve significant health benefits [36,37,38]. An analysis of 17 prospective cohort studies reported that every 2 g increase in cereal fibre intake per day was associated with a 6% reduction in the risk of developing type-2 diabetes [39], and a meta-analysis study reported that a 7 g/day increase in fibre intake can decrease the risk of cardiovascular disease, haemorrhagic and ischaemic stroke, diabetes, colorectal cancer, and rectal cancer [21].

Progress towards an accepted definition for prebiotics has spanned decades. First defined in 1995 as a “non-digestible food ingredient that beneficially affects the host by selectively stimulating growth and/or activity of one or a limited number of bacteria already resident in the colon” [40], this definition was updated in 2004 to “selectively fermented ingredients that allow specific changes, both in the composition and/or activity in the gastrointestinal microflora that confers benefits upon host well-being and health” [41]. Currently, the International Scientific Association for Probiotics and Prebiotics (ISAPP) consensus panel’s definition states that a prebiotic is “a substrate that is selectively utilized by host microorganisms conferring a health benefit” [42].

The prebiotic properties of fermentable fibres such as inulin-type fructans, galactans, and resistant starch (RS) have been the focus of much research [43,44,45,46,47,48]. When these fibres enter the colon, they are degraded by the gut bacteria to oligosaccharides or monosaccharides using a variety of carbohydrate active enzymes (CAZymes) and absorbed as an energy source through specific transport systems [49]. Symbiotic relationships between gut microbiota members, including for example representatives of *Bifidobacterium* and *Lachnospiraceae*, facilitate the optimal metabolism of fermentable intermediate molecules in a manner that is driven by the quantity, quality, and type of carbohydrate available [50]. Bifidobacteria can utilise a diverse range of polysaccharides [45], and representatives are often seen to increase in abundance due to interventions with prebiotic fibres. Corresponding keystone species in the gut can contribute to community symbiosis by providing energy to other residential microbial communities in the form of acetate and lactate as cross-feeding metabolites [45,51].

Interventions involving prebiotic fibres are mostly limited to isolated single fibres, combinations of two, or triads. In relative terms, there is a lack of literature on the prebiotic effect of a comprehensive mix of soluble and insoluble fibres in a food readily available to the consumer. It is also important to consider that fibres may function differently than when incorporated together in a food matrix, such as a beverage [15].Therefore, our study aimed to contribute to addressing these knowledge gaps by investigating the effect of a fruit juice drink containing a fibre blend consisting of chicory root inulin, green banana, golden kiwi fruit, and baobab, and thus including soluble dietary fibres, pectins and fructans, insoluble resistant starch 2 (RS2), and cellulose lignin on the gut microbiota structure and function, stool consistency, and emotional health in healthy humans.

## 2. Materials and Methods

### 2.1. Participants

#### 2.1.1. Recruitment Procedure

Study participants were recruited through social media and poster advertisements around London South Bank University campus. A total of thirty-five individuals expressed their interest in participating. After reading the participant information sheet, fourteen individuals reported they were not eligible to participate due to not meeting the eligibility criteria, or not being available for the whole study duration. Twenty-one participants were enrolled.

Inclusion criteria: to be eligible for the study, participants must self-report to be healthy; be over the age of 18; have a BMI of between 18.5 and 29.9; and live in the UK.

Exclusion criteria: not consumed prebiotic- or probiotic-containing foods, drinks, and supplements within the past 1 month; not be vegan or vegetarian; not experienced a significant change in weight (±5% of total body weight) or dietary intake over the past year; not taken antibiotics within the past 6 months; not pregnant or breast feeding; not a high alcohol consumer (>15 standard drinks per week for males or >10 standard drinks per week for females, and <2 days per week alcohol free); they or their immediate DNA family (mother, father, brother, or sister) have not had a medical diagnosis for ulcerative colitis, Crohn’s disease, irritable bowel syndrome, bowel cancer, or any inflammatory bowel disease; not be allergic to any of the ingredients in the product; not had brain cancer; no history of mental health disorders, or diagnosed with a disease or condition for which they receive NHS care/support/medication; not had a positive COVID-19 diagnosis, or displayed suspected symptoms in the last 6 months; and not on hormone replacement therapy or following a low-FODMAP diet. Employees of MOJU Ltd., and their family members, were not eligible to participate in this project.

A total of twenty eligible participants provided written informed consent to participate in this human intervention study (Figure 1). Three participants withdrew after providing baseline data due to product delivery delays impacting on their availability to complete the whole study. Three further participants were withdrawn during the study, two due to testing positive for COVID-19 and one due to starting a course of antibiotics. Fourteen healthy participants, all non-smokers (four male and ten female), completed the entire 9-week study period. One participant’s (TS2) stool sample for the placebo phase (treatment order prebiotic start) was lost in the post. Baseline characteristics (mean ± SD) of the 14 individuals who completed the study were: age 42.7 ± 12.4 (range 30–68); body weight 72.6 ± 13.3 kg (range 58–102); and BMI 24 ± 2.4 (range 22–28) (Appendix A).

The study was fully explained to the participants, in writing, and each gave their written, informed consent prior to participating.

#### 2.1.2. Randomisation and Group-Allocation Procedure

Sealed Envelope Ltd. [52] software was used to apply the block randomisation method to randomise participants into one of two intervention orders (i.e., AB-prebiotic then placebo or BA-placebo then prebiotic).

#### 2.1.3. Investigational Product

A raspberry, lemon, and baobab prebiotic shot was commercially produced by MOJU Ltd., London, UK, containing the following ingredients: apple, MOJU Prebiotic blend (chicory root inulin 4%, green banana powder, golden kiwi powder, and baobab powder 1%), lemon 8%, ginger root, raspberry, apple cider vinegar, and ascorbic acid. Participants consumed a 60 mL portion daily, providing 2.4 g of chicory inulin. The commercially available MOJU Prebiotic Shot (berry) is a single serving designed as a convenient, concentrated functional ‘shot’. The specific quantitative breakdown of the formulation is proprietary information. The placebo product contained the same ingredients, without the MOJU Prebiotic blend. Participants were recommended to consume their 60 mL juice shot first thing in the morning on an empty stomach. See Appendix A for nutritional data of the MOJU Prebiotic Shot.

#### 2.1.4. Study Design and Procedure

The study adopted a randomised controlled, cross-over design (Figure 1). The sample size was determined based on the findings from previous studies with prebiotic fibre interventions in healthy humans [53,54,55,56]. See Appendix A for the CONSORT (Consolidated Standards of Reporting Trials) flow diagram.

The researcher involved in participant recruitment and data collection was unblinded to the intervention order. The participants were blinded to the intervention order.

Participation in this study was entirely online. Prior to commencing the intervention (week 0), participants provided a stool sample, completed a depression, anxiety, and stress (DASS-42) questionnaire [57] completed a Bristol Stool Scale (BSS) chart rating their average weekly consistency for the previous week [58] and completed a fructan food frequency questionnaire (F-FFQ) based on the previous week’s dietary intake [59]. Participants received their allocated products for phase 1 via a courier. At the end of each intervention week (weeks 1–3) participants completed DASS-42, F-FFQ, and BSS questionnaires. In addition, at the end of phase 1 (week 3), participants provided a second stool sample. At the end of the 3-week washout period (week 6) participants provided a third stool sample and completed the three questionnaires again. Participants received their allocated products for phase 2 and completed the same questionnaires at the end of each week (weeks 7–9) and provided their fourth stool sample at the end of week 9. Participants received weekly emails containing the link to that week’s online questionnaires. If participants did not complete their questionnaires within 24 h of receiving the notification email, they received daily reminder emails as well as reminder text messages to their mobile phones. Participants were also sent an email enquiring about adverse symptoms at the end of each intervention week. Participants were sent four faecal sample kits (Zymo Research, Freiburg, Germany) in the post. Faecal samples were taken by the participants at home and returned via special postal delivery. Samples were then sent to SeqBiome Ltd., Cork, Ireland (Seqbiome.com) for shotgun metagenomic analysis.

### 2.2. Gut Microbiome Analysis

#### 2.2.1. Sample Collection and Analysis

Microbiome sample collection was performed using the DNA/RNA Shield-Fecal collection tube and Feces Catcher (Zymo Research, Europe).

DNA from samples was extracted with the QIAamp^®^ Fast DNA Stool Mini Kit (Qiagen, West Sussex, UK) using an adapted protocol. A total of 200 µL homogenised faecal sample was transferred to a screw-cap tube containing 1 mL of InhibitEX and a mixture of different-sized sterile zirconia beads (all beads provided by Stratech). Then a 3 min bead beating step was performed to lyse the cells using a TissueLyser II (Qiagen) before the tube was heated for 5 min at 95 °C. From this point, extraction was carried out according to the manufacturer’s instructions for “pathogen detection” (Qiagen, West Sussex, UK). DNA was eluted in 100 μL and stored at −20 °C until use in library preparation.

DNA was quantified using the Qubit double-stranded DNA (dsDNA) high-sensitivity assay kit (Bio-Sciences, Dublin, Ireland). All samples were prepared for shotgun metagenomic sequencing according to Illumina Nextera XT library preparation kit guidelines, with the use of combinatorial dual indexes (CDI) for multiplexing with the Nextera XT index kit (Illumina, San Diego, CA, US). Final clean libraries were assessed on the Agilent Bioanalyzer 2100 using a DNA High Sensitivity Assay Kit (Agilent, Santa Clara, CA, US), quantified by Qubit as before, and pooled using equimolar concentrations (15 nM). A final sequencing pool quality check and quantification was performed by qPCR using the KAPA Library Quantification Kit for Illumina (Roche KAPA, Basel, Switzerland). Sequencing was performed using a NextSeq 500/550 Mid Output Kit v2.5 (300 cycles).

Quality assessment of raw sequencing data was performed using *FastQC*, which allows quality visualization and manual quality inspection of raw sequence data [60]. All samples were of generally good quality with minimal quality drop-off at the end of sequences. All read statistics were relatively consistent, with on average ~76% of reads passing quality filtering steps and an average of ~86% of these being classified using *Kraken2* (see Appendix A). From this it was passed to *Trimmomatic* [61] for trimming and quality filtering using the following parameters: SLIDINGWINDOW:5:22 and MINLENGTH:75. The data were then passed to *Kneaddata* (https://huttenhower.sph.harvard.edu/kneaddata/) (accessed on 27 September 2022) for contaminant removal. Pathway and gene family assignment was then performed using *HUMAnN 3* [62]. The standard Uniref90 [63] annotation of gene families was collapsed into Kyoto Encyclopedia of Gene and Genomes (KEGG) orthology annotation and further collapsed into KEGG module annotation (https://www.genome.jp/kegg/ accessed on 27 September 2022). Taxonomy assignment was performed using *Kraken2*, a taxonomic classification system using exact k-mer matches to achieve high accuracy and fast classification speeds, using a customised version of the Genome Taxonomy Database (GTDB), the results of which were analysed using the R-based phyloseq package [64,65] GTDB clusters available genomes based on Average Nucleotide Identity (ANI) and assigns taxonomy based on the National Centre for Biotechnology Information (NCBI) (https://www.ncbi.nlm.nih.gov/ accessed on 27 September 2022) classifications. If a sequence does not meet clustering requirements (95% ANI), but does not have a unique taxonomic classification, a suffix will be added to indicate this (e.g., *Escherichia coli_A*). This allows for the most accurate taxonomic classifications, even in the case of organisms which have yet to be formally identified.

#### 2.2.2. KEGG Pathways and Modules

Gene families and metabolic pathways were generated using *HUMAnN3*, which is a system for accelerated functional profiling of shotgun metagenomic and meta transcriptomic sequencing from host- and environmentally associated microbial communities [62]. The generated gene families were then converted from Uniref90 annotation to KEGG modules. KEGG is a database of orthologs organised in a hierarchical fashion into pathways and high-level modules [66].

KEGG pathways and modules were assessed for differential abundance using *ALDEx2*. Using an uncorrected *p*-value of 0.05, KEGG pathways and KEGG modules were analysed by computing PCAs using Aitchison distances. To test the significance of groupings, PERMANOVA tests were performed on these distances, using 999 permutations, with a *p*-value cut-off of 0.05. Dispersion homogeneity was tested using the *betadisper* function from *vegan* [67].

### 2.3. Statistical Analysis

SPSS IBM V27.0 and R v4.1.2 software was used to analyse all the data, using the packages *vegan, CoDaSeq, Compositions, ggplot2, phyloseq*, and *ALDEx2*. Paired-sample t-tests (or the related samples sign test) were used to determine within-group differences, and independent-sample t-tests (or Mann–Whitney U test) were used to determine between-group differences. To account for the compositional structure of the Next Generation Sequencing (NGS) generated data and to avoid the likelihood of generating spurious correlations, we first inputted zeros in the abundance matrices using the count zero multiplicative replacement method (CMultRepl, method  =  “CZM”) implemented in the *Compositions* package and applied a centred log-ratio transformation (CLR) using the *codaSeq.clr* function in the *CoDaSeq* package. Alpha and beta diversity analysis was carried out using the R packages *vegan* and *phyloseq*, and statistical comparison was tested using Kruskal–Wallis tests. Principal coordinate analysis (PCA) was carried out using the *PCA* function in R using Aitchison distance matrix (CLR + Euclidean distances). The differences between species-level taxonomy for every treatment group at each time point was analysed using paired Wilcoxon tests provided by *ALDEx2*, using an uncorrected *p*-value of 0.05. Uncorrected *p*-values were used as no significance was observed using corrected *p*-values; thus these results should be interpreted with some caution.

*ALDEx2* was used to determine if any species or functional pathway modules were differentially abundant in any groups between timepoint or at any time point between the groups based on a Kruskal–Wallis test using 512 Monte-Carlo instances drawn from the Dirichlet distribution.

Estimation of alpha diversity was performed using Shannon [68], Chao 1 [69], and Simpson [70] diversity metrics. Some carryover effect was observed, and so to account for this, the difference/delta of alpha diversities between time points was analysed for significance using Wilcoxon tests.

Beta diversity was visualised by performing principal component analysis (PCA) using Aitchison distances. To test the significance of these groupings, a Permutational Multivariate Analysis of Variance (PERMANOVA) test was then performed using 999 permutations with a significance cut-off value of 0.05. Dispersion homogeneity was tested using the *betadisper* function from *vegan* [67]. DASS-42 data were analysed with a two-way repeated-measures ANOVA with Bonferroni correction for multiple comparisons. Differences were considered statistically significant at *p* ≤ 0.05.

## 3. Results and Discussion

### 3.1. Habitual Dietary Intake of Inulin and Oligo Fructose

Long-term dietary patterns and habitual food intake have significant influence in shaping individual stable gut microbiota structures, thought to explain over 50% and 20% of variability in mice and man, respectively [71]. Dietary fibre has consistently been demonstrated to impact the gut microbiome [72], and fermentable fibres in the habitual diet would result in a gut microbiome structure designed for optimal and efficient metabolism of these fibres. Therefore, to establish potential influences for variable responses to the MOJU Prebiotic Shot, we investigated the habitual dietary intake of inulin and oligo fructose in our cohort. Participants completed a validated Fructan-FFQ (F-FFQ) [59] questionnaire. For each food item, participants indicated their portion size (small, medium, or large) and the number of portions consumed over the previous 7 days. Portion sizes were estimated using NUTRITICS nutritional analysis software, Research Edition v5.82 (Nutritics.com). Inulin and oligo fructose amounts were calculated following the methodology previously performed [73]. Composite cereal products, such as breakfast cereals and cereal bars, were excluded from the calculation due to lack of detailed composite information and/or brand names. We therefore highlight the need to request this information in future studies.

There was a significant between-group difference in background dietary inulin intake during the placebo phase only (Table 1), with the treatment order placebo-start (BA) group consuming 2.43 g/day more (mean difference of 2.43 g/day, 95% CI, 4.73 to 0.137, *p* < 0.05, paired *t*-test) (see Appendix A for the prebiotic phase). However, when we pooled the data for the whole cohort, we observed no significant difference in habitual dietary intake of inulin between the prebiotic and the placebo phase, *p* > 0.05 (see Appendix A). There were no significant within-group differences in inulin and oligo fructose intakes (*p* > 0.05) between the placebo or the prebiotic phases (see Appendix A), nor between-group differences at baseline (*p* = 0.639 and 0.645, respectively; independent *t*-test).

Data on population average dietary intakes of naturally occurring inulin and oligo fructose were reported in 1999 for the US (2.6 and 2.5 g, respectively) and in 1995 for Europe (2–10 g combined inulin and oligo fructose) [74]. Recent data, however, remain elusive. In the present study we observed an average (mean ± SD, n = 14) daily intake for naturally occurring inulin and oligo fructose of 5.49 ± 2.5–6.07 ± 3.01 and 4.73 ± 2.34–5.12 ± 2.58 g/day, respectively (Appendix A). However, since we did not include composite cereal products, the average daily intakes may be higher than reported here, especially since both inulin and oligo fructose are frequently used in food products such as ice cream, yoghurt, cheese, and baked goods to replace sugar and fat [75,76].

### 3.2. Stool Consistency

There were no within-group significant changes in self-reported weekly stool consistency during the prebiotic fibre intervention in either treatment order group (*p* > 0.05) (see Appendix A). Both groups reported softer stools during the prebiotic and placebo phase, and one possible explanation for this is a strong placebo effect from participating in this study [77]. We did not collect data on stool frequency during this study. One participant in the AB group reported their stool frequency reduced during the prebiotic phase, but did not provide a number in reductions, and one participant, also in the AB group during the prebiotic phase, reported an increase in daily stool frequency from 1 to 2 per day for the first two weeks and from 2 to 3 per day during the third week. Prebiotic fibres seem to impact stool consistency and frequency in those with low frequency only [78].

### 3.3. Safety and Side Effects

One participant (AB group) noted increased flatulence and a ‘grumbling tummy’ during the first week only of consumption of the prebiotic drink. One participant (AB group) noted increased intestinal bloating for the first 3 days only, and one participant (AB group) experienced increased flatulence with a sulphuric smell across the whole prebiotic intervention. There were no reported side effects from any of the BA group members during the prebiotic intervention; one reported stomach pain after the first placebo shot on an empty stomach.

### 3.4. Depression, Anxiety, and Stress

We observed no significant changes in depression, anxiety, or stress within any of the groups during the prebiotic or placebo phases (see Appendix A). Several participants failed to submit their DASS-42 data on either side of an intervention phase (prebiotic or placebo), which reduced the number of data sets available for analysis to ≤5 per group. Previous studies have reported prebiotic fibre amelioration of stress [79], anxiety [80], and depression [81], whereas Johnstone and colleagues observed an improvement in highly anxious participants only [82]. Subject to a high number of reviews over the past 5 years, the conclusions vary between no evidence and very little evidence for prebiotic effects on depression and anxiety [83,84,85,86].

### 3.5. Microbiome Composition Changes Due to the Intervention

#### 3.5.1. Alpha Diversity Analysis

Both Shannon and Simpson diversity was observed to be significantly enriched as a result of prebiotic treatment when compared to placebo (*p* = 0.0027 and 0.0056, respectively) (Figure 2A,B). This is in agreement with the literature, as prebiotics have been shown to increase gut microbiome alpha diversity in a number of instances [87,88].

Multiple linear regression demonstrated that participant age and gender did not significantly influence response to the intervention for total relative abundance, *p* > 0.05 (see Appendix A).

#### 3.5.2. Beta Diversity Analysis

Treatment grouping/phase was found to be significant in regard to grouping across PCA dimension one (*p* = 0.001) and to be significant/borderline significant in regard to dispersion across PCA dimensions one to three (*p* = 0.0368, 0.0416, and 0.0624, respectively). This indicates that there is a considerable carryover effect, as samples tend to be grouped by treatment group and not treatment alone (Figure 3).

#### 3.5.3. Between-Intervention Differential Abundance Analysis

Differential abundance analyses of taxa aimed to detect if there were differences in composition between the prebiotic and placebo [89]. We observed no significant differences in the six main bacterial phyla between the two groups at baseline (Table 2); however, within-group responses were highly variable (Figure 4 and Table 2) and, similar to Ramirez-Farias et al. [90], we observed a treatment-order response. Only the participants in the prebiotic-start (AB) group responded significantly during the prebiotic phase with respect to *Desulfobacteria* (*p* = 0.016, unadjusted) and *Actinobacteria* (*p* = 0.016, unadjusted). The increase in *Actinobacteria*, the phylum that contains the genus *Bifidobacterium*, is consistent with results obtained by others [91], and substantial inter-variation responses to prebiotic fibres is frequently reported in reviews [28]. After a multiple testing adjustment using the Benjamini–Hochberg method, only the changes in the *Actinobacteria* remained significant (*p* = 0.048). *Actinobacteria* in the treatment order placebo-start (BA) group did not increase during the prebiotic phase. A bloom in *Desulfobacteria* from prebiotic fibres has been reported previously [92] (see Appendix A for stack bar charts of phylum level changes).

Nine species were observed to be differentially abundant between time points under prebiotic treatment (Figure 5); of these species, *Bifidobacterium adolescentis* (*p* = 0.03, unadjusted) and CAG-81 sp900066785 (*p* = 0.01, unadjusted) were increased relative to the baseline timepoint (Figure 5A). Three species were observed to be differentially abundant under placebo treatment, namely *Coprococcus_A catus* (*p* = 0.04, unadjusted), ER4 sp900317525 (*p* = 0.04, unadjusted), and CAG-83 sp000431575 (*p* = 0.03, unadjusted) (Figure 5B). *C. catus* decreased relative to baseline, ER4 sp900317525 showed a slight increase, and CAG-83 sp000431575 increased. However, none of these changes remained statistically significant after correction for multiple testing. Multiple linear regression demonstrated that participant age and gender did not significantly influence response to the intervention for both *B. adolescentis* and CAG-81SP900066785, *p* > 0.05 (see Appendix A).

Of these species, perhaps the most relevant is *Bifidobacterium adolescentis*, which shows a clear pattern of enrichment under prebiotic treatments (Figure 2C). This is in agreement with the literature, with *Bifidobacterium* species being one of the commonly enriched taxa in response to many prebiotic treatments [91], including pectin [93,94,95]. CAG-81 sp900066785, from the Firmicutes phyla, and the *Lachnospiraceae* family, is regularly observed in healthy humans with the average abundance detected ranging from 0.06% to 2.25% [96]. Members of this family are known carbohydrate fermenters and starch degraders [97] and are demonstrated to be increased by inulin and FOS supplementation in murine models [98,99] including pectic oligosaccharides (POSs) [93,95,100]. *Coprococcus catus*, which decreased during the placebo phase and is another member of the *Lachnospiraceae* family, is one of the core members of the human gut microbiota and is demonstrated to contain fructan and starch degradation enzymes. It is a succinate, acetate, lactate, and propionate producer, and was demonstrated to grow on chicory inulin and apple pectin in vitro [101], and was enriched in the gut microbiome, alongside *Bifidobacterium*, of pectin-fed rats [94]. It is therefore probable that the withdrawal of the prebiotic fibres during the placebo phase impacted on *C. catus* proliferation.

CAG-83 sp000431575 is from the Firmicutes phylum, family *Oscillospiraceae*, and was demonstrated by the same group to be enriched in the gut microbiome in pectin-fed rats [94]. *Oscillospira* is considered one of the next-generation probiotic candidates. ER4 sp900317525 is also from the family *Oscillospiraceae*. They are butyrate producers and ferment complex plant carbohydrates [102]. Moreover, bifidobacteria [103], FOS [104], and an increase in plant-based meat alternatives in the diet known to contain higher levels of mixed dietary fibres [105] have demonstrated to have a positive effect on *Oscillospira*. An increase observed in these two probiotics during the placebo phase in our study could indicate a carryover effect. The precise definition of what constitutes a carryover effect is not clear, and we propose that the increase in CAG-83 sp000431575 and ER4 sp900317525 during the placebo phase is an example of ‘prolonged benefit.’

Notable individual variable responses include subject TS2, who displayed a large shift in taxonomic composition between time points during prebiotic treatment, and subject VI4, who showed high levels of *Akkermansia* during the placebo treatment but not during prebiotic treatment (Figure 4).

#### 3.5.4. Microbiome Functional Changes Due to the Intervention

##### Metagenome Functional Analysis

Several significantly differentially abundant pathways and modules were observed between time points during both placebo and prebiotic treatment, again indicating a potential carryover effect (Figure 6). The KEGG pathway associated with the citrate cycle was decreased during both placebo (*p* = 0.03, unadjusted) and prebiotic (*p* = 0.02, unadjusted) treatment. The pathway associated with vitamin B6 metabolism (*p* = 0.04, unadjusted) was downregulated during prebiotic treatment, and the pathways associated with microRNAs in cancer (*p* = 0.01, unadjusted) and cysteine/methionine (*p* = 0.04, unadjusted) were decreased during placebo treatment. In a similar fashion, multiple KEGG modules associated with the citrate cycle (Figure 6) were decreased during both treatments (prebiotic *p* = 0.05; placebo *p* = 0.04, unadjusted). Perhaps of most interest is the various upregulated modules associated with arginine metabolism observed during the prebiotic treatment (*p* < 0.05, unadjusted) (Figure 6A and Figure 7). None of these changes remained statistically significant after correction for multiple testing.

The presence of a carryover effect is further supported by the fact that during PCA analysis, treatment grouping was significant in regard to grouping across all three PCA dimensions, and significant in regard to dispersion across PCA dimensions one and two. While treatment was also observed to be significant in regard to grouping and dispersion across multiple dimensions, it seems that much of this variability may be due to treatment grouping, making it difficult to determine if this significance is genuine. This can be seen in Figure 7, where the most distinct clusters consist nearly solely of samples from the prebiotic start group during placebo treatment, i.e., the samples from the longest time after prebiotic treatment. This may indicate that there is a longer-term effect of prebiotic treatment which is not completely accounted for by the 21-day washout period. This pattern is consistent between both KEGG pathways and KEGG modules.

## 4. Discussion

In this study we investigated the effects of a fruit juice drink (MOJU Prebiotic Shot) containing a mixture of fermentable and prebiotic fibres on the structure and function of bacterial communities in the human colon, depression, anxiety and stress, and stool consistency, utilising a cross-over approach.

The fermentable fibres in the MOJU Prebiotic Shot include non-starch polysaccharides, specifically chicory inulin and pectic-oligosaccharides from golden kiwi fruit and baobab, as well as resistant starch (RS2) from green banana. These fibres are resistant to host enzyme degradation and pass through the small intestine into the colon where they can be fermented by a variety of gut commensals [106]. For a negative control, we used the same fruit juice drink that did not contain the mixture of fibres. A 60 mL portion contains 3.7 g of dietary fibre, 2.4 g of which is contributed by chicory root inulin.

Inulin, considered the ‘gold-standard’ prebiotic, is the most extensively studied prebiotic fibre to date [46,107]. Commercial baobab fruit pulp powder is rich in polyphenols and flavonoids and contains around 45–54% (dry weight) fibre, 50–75% of which is soluble fermentable fibres, including pectin-based polysaccharides (PPOSs) and oligosaccharides (POSs) (42.5%) [108,109]. POS has previously been demonstrated to have prebiotic effects [110], including bifidogenic effects [109,111,112,113].

Green banana flour (GBF) contains around 8.5–15.5g/100 g of fibre, with a soluble/insoluble fibre ratio of 1:5, and 40–54.2 g/100 g resistant starch (RS2) [15,114,115,116]. Li et al. [115] demonstrated that GBF restored gut permeability and intestinal barrier function over a 2-week intervention in an antibiotic gut microbiota dysbiosis model in rats faster than natural recovery, by increasing mucin secretion. Various beneficial effects of GBF have been reported in animal models, including modulating intestinal inflammation in a colitis model [117], improvement of obesity parameters [114], increasing SCFAs, and modulating oxidative stress [118].

Kiwi fruit is rich in POS, polyphenols, and flavonoids. Both green and gold varieties have been investigated for their gastrointestinal (GI) symptom improvements in humans [119,120,121] and pigs [122]. Using an in vitro digestion model, Parkar and colleagues demonstrated that gold kiwi fruit increased *Bifidobacterium*, including enhancing the adhesion to intestinal epithelial cells [123].

The MOJU Prebiotic Shot used in our study elicited a significant change in the overall composition of the gut microbiome, as demonstrated by both Shannon and Simpson diversity indices. Alpha diversity changes from prebiotic fibre interventions are not consistently reported, and great variability exists amongst those who do report results [124]. Fibre dosage does not appear to influence alpha diversity; for example, chicory inulin at 20 g and 10 g [78] per day produced no observable changes, whereas 12 g [125] and 6 g/8 g [42] per day produced reported reductions in alpha diversity. One of the reasons for such variability may be due to the sequencing method used. General 16S rRNA amplicon (V3-V4) sequencing can produce false positive results, overestimating certain taxa [71] and underreporting alpha diversity scores [126], relative to shotgun sequencing [1,127], which would increase alpha diversity scores when compared with short-read 16S rRNA sequencing.

We observed grouping of paired samples (beta diversity) according to treatment order rather than treatment phase, indicating potential of a carryover effect where the 3-week washout period in the treatment order prebiotic start (AB group) was not sufficient to return the gut microbiota composition to pre-intervention baseline. Fibre intervention trial design was reported to influence the magnitude of outcomes with parallel design studies demonstrating stronger intervention effects and greater statistical heterogeneity in comparison to a cross-over design, seemingly for several outcomes, and the carryover effect is proposed to be one of the reasons [124,128]. Reviews on fibre interventions have reported that not all cross-over trials include washout periods [20,124] and reporting of a carryover effect after a washout period is rare. It is mainly accepted that fibre-induced changes to the gut microbiota are only maintained whilst the fibres are consumed [37,71]. We chose a 3-week washout period based on expert recommendations and previous findings [53,129,130,131,132] including practical considerations such as duration of participant commitment and controlling the influence of seasonal and holiday dietary changes. Furthermore, although we observed no significant differences between groups in habitual dietary inulin and FOS levels at baseline, and pooled data demonstrated no significant difference between the intervention phases, we did observe a between-group difference (*p* < 0.05) for inulin during the placebo phase, and the potential contribution of this difference to our treatment order groupings should not be ignored. It has been proposed to stratify participants according to their baseline microbiota enterotype and include an initial exploratory study to investigate the required washout period [133]; however, beyond the cost implications and practical feasibility, it would divert from the purpose of our study which was to investigate the effect of a commercially available food product in the average healthy consumer.

Similar to other studies investigating gut microbiota changes in humans from prebiotic fibres, we observed a bifidogenic effect [55,56,90,130,134,135,136], in combination with an increase in *Lachnospiraceae* [98,137]. *B. adolescentis* appears to respond selectively to inulin [28,90,130,138,139,140,141] and resistant starch [37,98,134,142,143]. Bifidobacteria in the gut are linked with numerous health benefits including SCFA production [144] and improved function of the gut barrier [145]. Reduced intestinal inflammation [146], a reduction in circulating lipopolysaccharides [147], and healthy gut immune responses [148] are all linked to improved gut barrier functionality. As keystone species, *Bifidobacterium* spp. can apply several survival strategies in a highly competitive environment, including glycan harvesting and glycan breakdown, leading to cross-feeding metabolites that support community stability [149]. Several studies have reported the symbiotic relationship between bifidobacteria and other members of the healthy gut microbiota by means of in vitro co-culturing experiments [97,150,151] including *B. adolescentis* [152]. The trophic interaction between bifidobacteria and *Lachnospiraceae* involving lactate as a cross-feeding intermediary metabolite has previously been reported [98,153,154] and is of great importance to the host and the gut microbiota community [155]. As one of the largest consortia of butyrate producers, we considered an increase observed in *Lachnospiraceae* to be a butyrogenic effect [153]. It is thought, in line with other studies [93,156,157] that the prebiotic fibres elicited an increase in butyrate production. An elegant study by Swanson and colleagues [28] gave insight into the cross-feeding and butyrate effect of chicory inulin in the human colon. They demonstrated that inulin fermentation was associated with a concurrent increase in butyrate, even though acetate and lactate are the only bifidobacterial fermentation products.

Furthermore, *B. adolescentis* is recognised for its increased genome stability, making it particularly competitive in the gut environment. In comparison to other bifidobacteria, *B. adolescentis* has more genes involved in quorum sensing, biofilm formation, the two-component system, and various carbon source and amino acid metabolism pathways, demonstrating its competitive advantage [158]. Indeed, several of the increased functional genes observed in our study, discussed below, are expressed by *B. adolescentis*, demonstrating its comprehensive capability of host and gut bacterial community health promotion.

We were particularly interested in the several upregulated amino acid (AA) related functional modules during the prebiotic treatment (Figure 6A). Two pathways for arginine biosynthesis (KEGG modules M00845 and M00844), the urea cycle and phenylacetate degradation to acetyl-CoA/succinyl-CoA *B. adolescentis*, express the genes for all these pathways (https://www.genome.jp accessed on 1 January 2023) and may be the reason for the upregulation of these functions observed in our study (Figure 8).

A variety of AAs are required for gut microbial production of SCFAs [159], microbial-derived neurotransmitters (e.g., GABA, dopamine, and serotonin), and, most interestingly, for amino-acid-dependent environmental stress-adaptation mechanisms [160]. Evidence for the *de novo* synthesis of some of the nutritionally essential amino acids [161] by gut bacteria, such as arginine, means that AAs can function as important microbe–host and microbe–microbe cross-feeding metabolites. For example, arginine, as a precursor for the biosynthesis of proteins and arginine-derived cross-feeding metabolites, including polyamines, nitric oxide, glutamate, urea, and creatine [24,162], is essential for the maintenance of intestinal microbiota homeostasis [163]. In the gastrointestinal tract (GIT) it drives three predominant metabolic pathways: (i) synthesis of polyamines through the provision of intermediate ornithine and agmatine; (ii) synthesis of urea, creatine, and ornithine; and (iii) direct catabolism into nitric oxide [159]. Competition for arginine in the GIT is high, since it is also extensively metabolised by intestinal epithelial cells (IECs) to produce adenosine triphosphate (ATP), mucins, immunoglobins, and defensins. Consequently, around 38–40% of dietary arginine in the intestinal lumen is catabolised in the first-pass order [164,165,166,167] highlighting the fundamental importance of the *de novo* synthesis of arginine in the colon by *B. adolescentis* to both host and gut microbiome health. Arginine supplementation has been demonstrated to influence gut microbiota composition in mice [49,168] and chickens [169]. Kim and colleagues reported an increase in *Bifidobacterium* spp. in mice [24], and Van Den Abbeele and colleagues [170] observed a significant increase in *Lachnospiraceae* in pigs. Taking the above into consideration, arginine functions as an important cross-feeding metabolite for beneficial gut bacteria.

Furthermore, arginine fulfils an important role in survival strategies activated in the gut microbiota under acid stress [160]. *Bifidobacterium* can significantly upregulate genes to support growth at low pH [160] and appears to use two different acid-tolerance response (ATR) protective mechanisms against acid stress. The F1-F0 ATPase [171] and the amino-acid dependent decarboxylase/antiporter system [172] both require arginine [160,173,174]. *B. adolescentis* is not acid tolerant and relies on these ATRs for survival and maintenance [171].

*B. adolescentis* is known to produce acetate and lactate as intermediate fermentation products [175], and increased colonic SCFAs from prebiotic fibre interventions are well reported [20,176], meaning that the MOJU Prebiotic Shot used in our study may have resulted in an increase in the colonic SCFA pool, and the resultant increase in intestinal acidity would have triggered a requirement for an ATR [177,178,179]. This may be one of the reasons why we observed the upregulation of two arginine biosynthesis pathways, as well as the urea cycle by *B. adolescentis* since urea is also utilised in an acid-tolerance response in gram-positive bacteria [160]. Ureolytic gut commensals, such as *Bifidobacteria* spp. [180,181], utilise urease [182] to turn urea into ammonia and carbonic acid, which increases intestinal pH, protecting intestinal bacteria from acidity that is either self-created (for example, from acetate or lactate producers such as *B. adolesentis*) or host-derived (for example, stomach acid) [183]. The ammonia produced here can enter the urea cycle as a substrate to produce arginine, urea, and ornithine, and ornithine can be used as an intermediate in the arginine biosynthesis pathway, leading to a perpetual recycling of important intermediates into arginine (Figure 8). Zarei and colleagues [184] demonstrated increased urea in the colon of germ-free (GF) mice from arginine metabolism through the urea cycle in comparison to conventionally raised mice, meaning that the gut microbiota is essential for urea catabolism. It is suggested that the microbial community can hydrolyse between 15 and 30% of urea synthesised in healthy subjects [185]. Urea, synthesised from arginine in the urea cycle, can be utilised for the de novo synthesis of microbial proteins; considering the importance of urease in nitrogen recycling, especially when diets are deficient in protein, makes the ability of the gut microbiome to produce and utilise urea, particularly advantageous to both microbe and host [185].

Arginine, therefore, can be seen as an important commensal gut microbiota currency, and the fact that the MOJU Prebiotic Shot elicited an increase in its production by *B. adolescentis* can be considered beneficial to both gut microbiota and host.

Furthermore, the ability of *B. adolescentis* to degrade colonic phenylacetate to the intermediate metabolites acetyl-CoA and succinyl-CoA is advantageous to both host and gut microbiota community. After carbohydrates, natural-occurring aromatic compounds, found as lignin (aromatic polysaccharide), flavonoids, quinones, and aromatic amino acids, are the second most abundant compounds in nature. Not surprisingly, these compounds can serve as growth substrates for microorganisms [186]. After cellulose, lignin is the most abundant polymer in nature [187]. Lignin, as part of the structural make up of plant cell walls, exists as lignocellulose, a complex mixture of carbohydrate polymers, hemicellulose, and cellulose covalently bound to lignin and pectic substance [188,189], and accounts for 30–50% of dry matter [190]. The cross-linked structure of hemicellulose and lignin polyphenols makes it more resistant to intestinal microbiota [191].

The fibres in the MOJU Prebiotic Shot used in our study are recognised sources of lignin. Baobab fruit cells (dried) contain 54% [192], and kiwi fruit (wet) contain between 8 and 12% [193]. Elsewhere, green banana was demonstrated to be a source of lignin [194]. Plants utilise the aromatic amino acids phenylalanine and tyrosine for the biosynthesis of phenolic compounds and lignin [195], and phenylacetate is a reductive intermediate of the anaerobic catabolism of these aromatic amino acids. Bifidobacteria was demonstrated to ferment lignin and metabolise the released phenolic compounds in vitro [196], and lignin–carbohydrate complexes have demonstrated to be metabolised by intestinal microorganisms to SCFAs and increase bifidobacteria [61,191].

Moreover, phenylacetate may also trigger virulence in pathogens [197], and high levels of intestinal phenylacetate have been demonstrated to be implicated in colorectal cancer [198] and autism spectrum disorder [199]. Consequently, the removal of circulating phenylacetate in the colon by *B. adolesentis* functions as an important step to host health maintenance and the prevention of pathogen proliferation in the gut.

Furthermore, the end products of phenylacetate degradation by *B. adolesentis*, namely acetyl-CoA and succinyl-CoA, function as cross-feeding metabolites to butyrogenic bacteria such as *Lachnospiraceae*. Both succinyl-CoA and acetyl-CoA are used to produce butyrate by butyrogenic bacteria [155], with the acetyl-CoA pathway demonstrated to be the most prevalent in mice [200]. *Lachnospiraceae* utilises the acetyl-CoA pathway to convert acetate to butyrate [201], and this cross-feeding example may be one of the reasons why studies with prebiotic fibres, such as ours, often report an increase in *Lachnospiraceae* alongside *Bifidobacteria* spp.

In addition, we observed an increase in *Desulfobacteria* (*p* = < 0.05, unadjusted) in one of our treatment order groups during the prebiotic intervention. Lignin and RS2 are both insoluble fibres, and Jangid and colleagues [92] recently demonstrated an increase in *Desulfovibrio* from an insoluble fibre intervention in mice, the bloom, hypothesised to be the result of increased hydrogen (H_2_) formation from an influx of insoluble fibres available for fermentation. Key bacterial fibre fermentation products include both SCFAs and gases such as hydrogen (H_2_) and methane [202], and members of the genus *Desulfovibrio* help in the removal of excessive H_2_ by converting it into hydrogen sulphide (H_2_S) frequently reported to have an odour like ‘eggy sulphur’, as reported by one of our subjects during the prebiotic phase in our study [203].

In addition, the prebiotic effect of the MOJU Prebiotic Shot used in our study is not limited to just the fermentable fibres. It contains a variety of polyphenols such as flavonoids and anthocyanins, considered to be substances with prebiotic activity [42,91,204] that have demonstrated a bifidogenic effect, modulate mucin synthesis, improve intestinal barrier structure and function, and increase SCFAs in vitro, in animals and humans [10,205,206,207,208,209,210]. It is proposed that the action of polyphenols on the gut microbiota relies on dual antimicrobial [211,212,213] and growth-stimulating effects [214,215] and has coined the term ‘duplibiotic’ [61]. Moreover, the effects of polyphenols on *B. adolesentis* have been investigated in vitro and demonstrated to promote growth [216], enhance anti-inflammatory activity, and stimulate production of acetate and lactate [217]. Furthermore, raspberries, rich in polyphenols such as ellagitannins and anthocyanins [218,219], increased *Lachnospiraceae* in an obese murine model [220], and ginger demonstrated a bifidogenic effect and increased SCFAs in mice [221] and alpha diversity in humans [222]. It is important to note that the placebo product used in our study also contained raspberry and ginger; the only difference between the MOJU Prebiotic Shot and placebo was the fibres.

Interestingly, the synergistic relationship between fibres and polyphenols have been reported [206,223,224,225]. Unless chemically extracted, polyphenols are intimately bound through hydrophobic and hydrogen bonds to plant cell wall matrix, such as lignin cellulose, and reach the colon in the form of polyphenolic fibres [226,227]. The fibre acts as an entrapping matrix and restricts host digestive enzyme access, meaning around 80–95% [228,229,230] of the polyphenols bound to these fibres reach the colon. Once the colonic microbiota ferments the fibres, resulting in SCFAs, the released polyphenols are deconjugated into microbiota-derived polyphenol metabolites [10,231] by members of the gut microbiota, such as *Bifidobacteria* and *Lachnospiraceae* [206,232,233] that express polyphenol-associated enzymes (PAZymes) [44]. In return, SCFAs facilitate the increased absorption of these metabolites [234] demonstrating a gut microbiota–fibre–polyphenol axis that is beneficial to the host [235,236,237]. This could be one of the reasons why conflicting results of the effects of polyphenols on gut microbiota are often reported since extracted polyphenols are not fibre-bound.

Taking the above into account, the phytochemicals present in the functional fruit juice drink with prebiotic fibres may have contributed to the structural changes and increased functional potential of the gut microbiota observed in our study in several ways: (i) directly by means of their proliferation effects on *Lachnospiraceae* and *B. adolesentis*, (ii) cross-feeding effects from the microbial phenolic metabolites, (iii) indirectly through their anti-microbial actions, and (iv) their beneficial impact on the mucosal epithelial barrier. It is important to note that we did not investigate the polyphenol content of the participant’s habitual diet and therefore cannot rule out a potential influence on the response.

We consider the cross-over design of our study a strength. The cross-over design of the study increases the power to detect changes in response to the treatment with a small number of subjects and a shorter period [135,238]; however, the design is also responsible for the main weakness of our study, which is considered to be the small sample size available for emotional health data processing. Limited funding prevented recruiting a number that took into account standard potential losses to follow-up, and losing three participants at the start due to product delivery issues was not anticipated. The online-only approach was both beneficial and detrimental. Finding participants who met the inclusion criteria and were able to commit to the duration of the study was difficult; reducing the need to travel to London South Bank University improved interest in participating. Conversely, ensuring participant compliance by completing the online questionnaires at the end of each intervention week proved a challenge and resulted in the greatest loss of data available for the depression, stress, and anxiety analysis.

Our results indicate that the MOJU Prebiotic Shot is safe for both men and women and that the daily consumption in healthy individuals provides a plausible amount of ingredients to demonstrate a prebiotic effect. Moreover, the provision of an additional 3.66 g of dietary fibre in a convenient manner means it may be beneficial to the general public, especially in terms of the increase in the relative abundance of *B. adolescentis* and CAG-81 sp900066785, a member of the *Lachnospiraceae* phyla, well known for their beneficial health effects.

To our knowledge, this is the first study providing insight into the ‘niche factor’ [239] strategies and functional potential of *B. adolescentis* brought about by a 3-week intervention of a fruit juice drink with prebiotic fibres in healthy humans. Our results provide evidence of complex cross-feeding pathways and indicate areas for future research such as the importance of the gut microbiota–arginine–host axis. In addition, we add to the lacking pool of data of fibre intervention trial washout duration and cross-over implications and the importance of measuring habitual dietary prebiotic fibres.

## 5. Conclusions

Three weeks of supplementation with 60 mL of the MOJU Prebiotic Shot, a fruit juice drink with a mixture of prebiotic fibres rich in secondary metabolites including polyphenols, significantly changed gut microbiota alpha and beta diversity, and resulted in measurable increases in beneficial organisms such as *B. adolesentis* and CAG-81 sp900066785, a member of the *Lachnospiraceae* phyla, in generally healthy individuals. Our metagenomic data revealed upregulated pathways and modules during the prebiotic intervention phase, shedding light on the specific functions assigned to *B. adolescentis*. In a world where we are not consuming enough dietary fibre, together with the knowledge that a variety of fibres in the diet matter, and that environmental influences to fibre content in food crops mean there is disparity between what we eat and what we think we eat [34], our results demonstrate the tangible benefit of a convenient daily fibre top-up in the form of a functional fruit juice drink with a mixture of prebiotic fibres.

## Figures and Tables

**Figure 1 foods-12-02480-f001:**
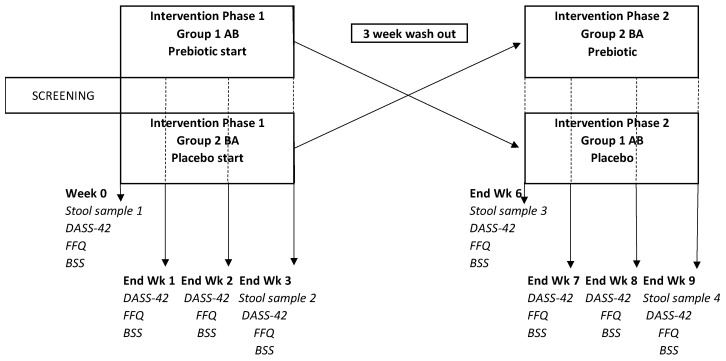
Study design.

**Figure 2 foods-12-02480-f002:**
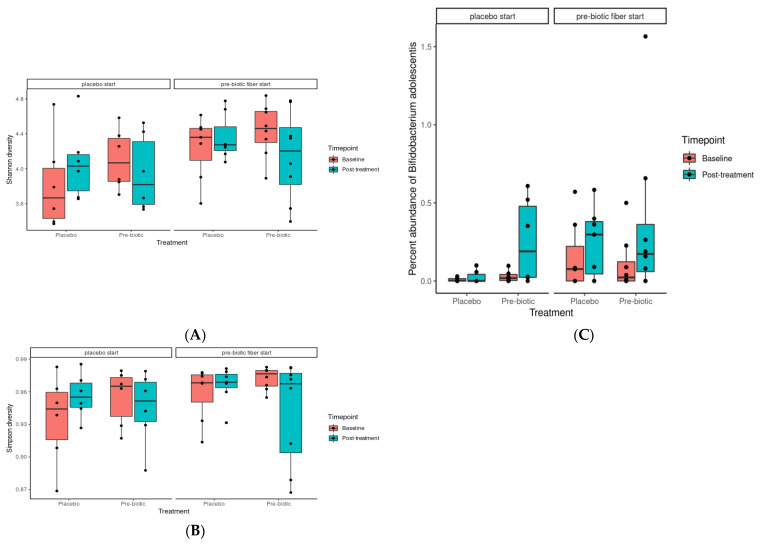
Effects of 3-week supplementation of the MOJU Prebiotic Shot or placebo (without fibres) on gut microbiota in healthy adults. Boxplot showing the (**A**) Shannon and (**B**) Simpson alpha diversity, grouped by treatment, coloured by timepoint, and faceted by phase. (**C**) Boxplot showing the percent abundance of *B. adolescentis*, coloured by treatment, grouped by time point, and faceted by treatment phase.

**Figure 3 foods-12-02480-f003:**
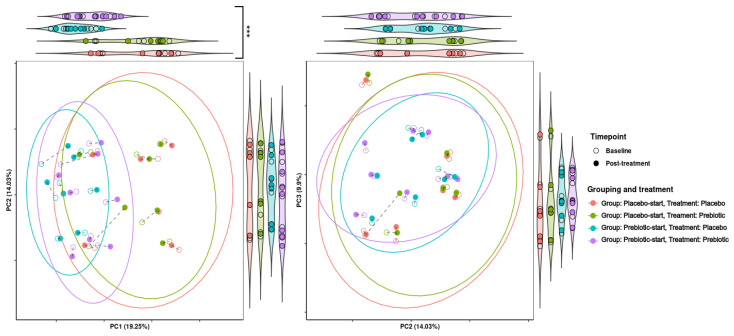
Effects of 3-week supplementation of the MOJU Prebiotic Shot or placebo (without fibres) on gut microbiota beta diversity in healthy adults. Principal component analysis (PCA) is based on Aitchison distance matrixes. PCA plots of PCA dimensions 1, 2, and 3 of beta diversity analysis. The ellipses represent 80% confidence intervals. Lines connect the same samples during each time point, with shapes indicating the time point, coloured by treatment and grouping. Both graphs are bordered by violin plots showing projections onto the principal component axes. Violin plots with significantly different groupings are marked, with *** = *p*-value < 0.01.

**Figure 4 foods-12-02480-f004:**
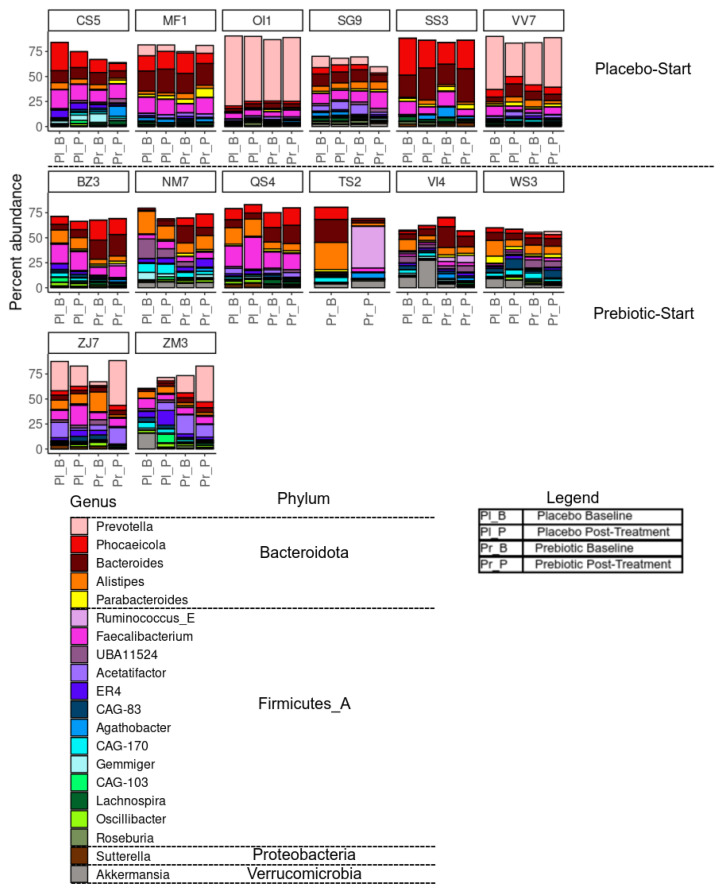
Effects of 3-week supplementation of the MOJU Prebiotic Shot or placebo (without fibres) on gut microbiota in healthy adults. Bar chart showing the percentage composition of the 20 most abundant species, coloured by genus, grouped by phylum, faceted by subject, and ordered by treatment and timepoint. Note: Firmicutes_A is a placeholder phylum which has been classified as Firmicutes in the NCBI database but does not meet the clustering requirements for GTDB. Unused space represents other species not in the top-20 most abundant.

**Figure 5 foods-12-02480-f005:**
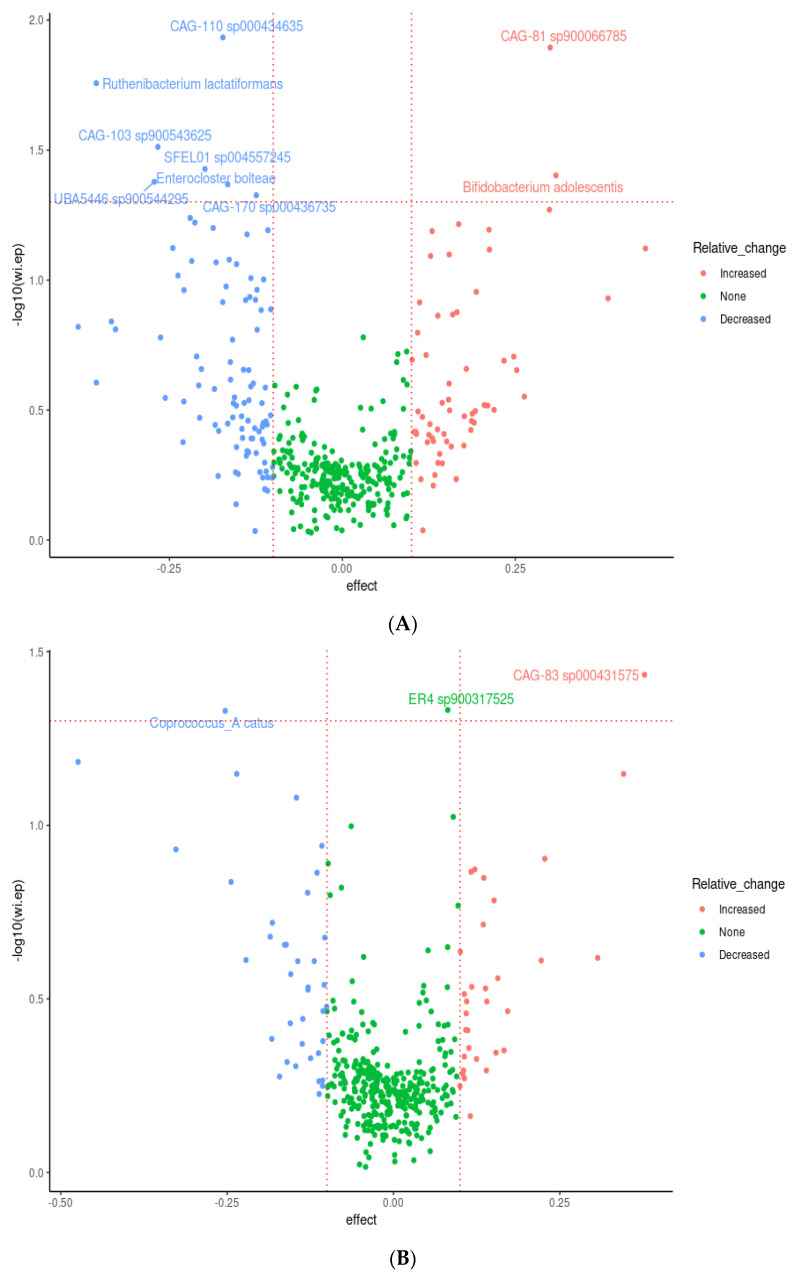
Effects of 3-week supplementation of the MOJU Prebiotic Shot or placebo (without fibres) on gut microbiota composition in healthy adults. Data represent a volcano plot of differential abundant gut microbiota species showing effect size and −log10 of the uncorrected *p*-value taxa between time points under (**A**) prebiotic fibres and (**B**) placebo (without fibres). Coloured by change relative to baseline and species with an uncorrected *p*-value < 0.05 labelled. The horizontal line represents the unadjusted *p*-value cut-off at 0.05. N = 14.

**Figure 6 foods-12-02480-f006:**
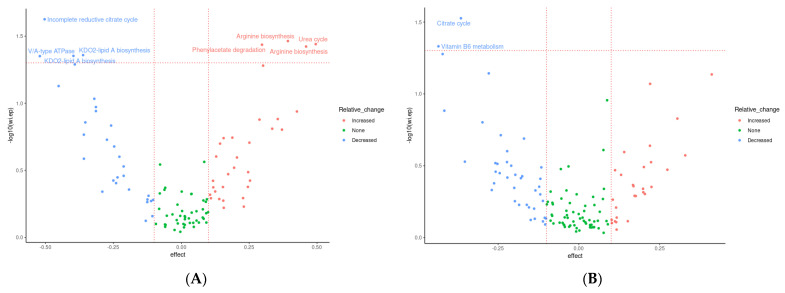
Effects of 3-week supplementation of the MOJU Prebiotic Shot or placebo (without fibres) on gut microbiota functional metagenome in healthy adults. Data represent a volcano plot of differential abundant KEGG modules (**A**) with prebiotic fibres and (**C**) placebo, and KEGG Pathways (**B**) with prebiotic fibres and (**D**) placebo, between groups. The *X*-axis position of each point represents effect size differences at the end of the intervention, coloured by change relative to baseline. The horizontal line represents the unadjusted *p*-value cut-off at 0.05; the vertical lines represent effect sizes of −0.1 and 0.1. N = 14.

**Figure 7 foods-12-02480-f007:**
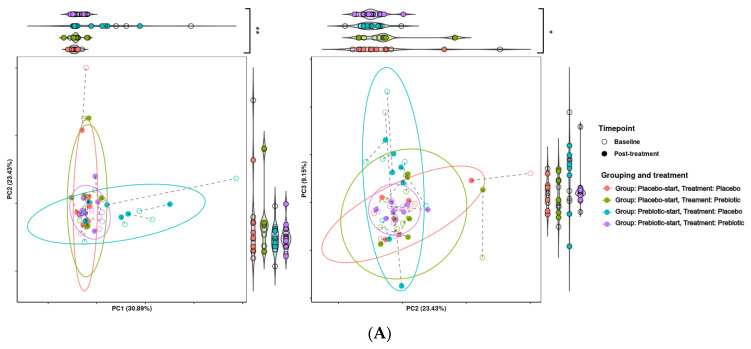
Effects of 3-week supplementation of the MOJU Prebiotic Shot or placebo (without fibres) on gut microbiota functional metagenome in healthy adults. Principal component analysis (PCA) is based on Aitchison distance matrixes. PCA plots of PCA dimensions 1, 2, and 3 of (**A**) KEGG modules and (**B**) KEGG pathways. Ellipses represent 80% confidence intervals. Lines connect the same samples during each time point, with shapes indicating the time point, coloured by treatment and grouping. Both graphs are bordered by violin plots. Violin plots with significantly different groupings are marked, with * = *p*-value < 0.1, ** = *p*-value < 0.05, and *** = *p*-value < 0.01.

**Figure 8 foods-12-02480-f008:**
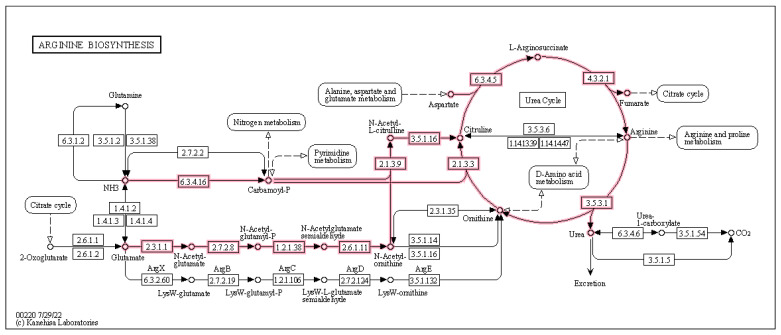
Effect of 3-week supplementation of the MOJU Prebiotic Shot or placebo (without fibres) on gut microbiota functional metagenome in healthy adults, showing upregulated arginine biosynthesis by *Bifidobacterium adolescentis* KEGG modules M00844, M00845, and the urea cycle M00029 during the prebiotic phase.

**Table 1 foods-12-02480-t001:** Habitual dietary inulin and oligo fructose during the placebo supplement phase. Data are in grams per day (mean ± SD); independent *t*-test.

Group/Treatment Order	Phase	Inulin	Between-Group *p*-Value	Oligo Fructose	Between-Group *p*-Value
AB	2	4.0 ± 1.65	0.04 *	3.47 ± 1.28	0.78
BA	1	6.44 ± 2.1		5.62 ± 2.52	

* Two-tailed.

**Table 2 foods-12-02480-t002:** Effects of 3-week supplementation of the MOJU Prebiotic Shot or placebo (without fibres) on individual responses of gut microbiota at genus level in healthy adults. AB is treatment order prebiotic start, and BA is treatment order placebo start.

Treatment Order	AB Prebiotic Phase		AB Placebo Phase		BA Placebo Phase		BA Prebiotic Phase		Baseline between-Group Differences (*p*-Values)
Results	Pre–post *p*-Value	Individual Responses N = 8	Pre–post *p*-value	Individual Responses N = 7	Pre–post *p*-Value	Individual Responses N = 6	Pre–post *p*-value	Individual Responses N = 6	
**Verrucomicrobiota**	0.125 **0.25**	7 ↑, 1 ↓	0.453 **1.0**	5 ↑, 2 ↓	0.062 **0.372**	5 ↑, 1 ↔	1.0	3 ↑, 2↓, 1 ↔	0.181
**Proteobactera**	0.453 **0.67**	5 ↑, 3 ↓	0.453 **0.9**	5 ↑, 2 ↓	0.688 **1.0**	4 ↑, 2 ↓	0.688 **1.0**	4 ↑, 2 ↓	0.534
**Firmicutes**	1.0	4 ↑, 4 ↓	1.0	3 ↑, 4 ↓	0.688 **1.0**	2 ↑, 4 ↓	1.0	3 ↑, 3 ↓	0.534
**Desulfobacteria**	0.016 * **0.09**	8 ↑	0.453 **0.67**	5 ↑, 2 ↓	1.0	3 ↑, 3 ↓	0.688 **1.0**	4 ↑, 2 ↓	0.628
**Bacteroidetes**	0.453 **0.54**	2 ↑, 6 ↓	0.125 **0.75**	1 ↑, 6 ↓	0.219 **0.65**	1 ↑, 5 ↓	0.688 **1.0**	4 ↑, 2 ↓	0.101
**Actinobacteria**	0.016 * **0.048 ***	8 ↑	0.453 **0.54**	5 ↑, 2 ↓	1.0	3 ↑, 3 ↓	1.0	3 ↑, 3 ↓	0.366

* Denotes significant *p*-value; related-samples sign test; Benjamini–Hochberg-adjusted *p*-values in **bold**; baseline between-group differences were analysed using the Mann–Whitney U test; ↑ = increase, ↓ = decrease, and ↔ = no change.

## Data Availability

The data used to support the findings of this study can be made available by the corresponding author upon request.

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
