# Peer review of "Shotgun Metagenomic Sequencing Revealed the Prebiotic Potential of a Fruit Juice Drink with Fermentable Fibres in Healthy Humans"

_foods, 2023, doi:10.3390/foods12132480_

Round 1
Reviewer 1 Report
The manuscript “Shotgun metagenomic sequencing revealed the prebiotic potential of a fruit juice drink with fermentable fibers in healthy humans”. Firstly, the novelty is not presented in the manuscript. Secondly, the number of participants for the test could be limited, and the age had a significant difference, leading to the result and conclusion could be uncertainty to some extent. Thirdly, the microbial community structure of these participants should be compared before and after test. In addition, the authors cited too many references, are them necessary? If the authors have to cite these references, please put the reference in the place that you cited, not be in together. The tables and figures are too casual.
Moderate editing of English language
Author Response
Dear reviewer, thank you for your valuable feedback. Please find the response to your comments below.
"Firstly, the novelty is not presented in the manuscript." We believe the novelty is strongly presented throughout the manuscript, starting with use of the words 'shotgun metagenomic' in the title; describing the gap in the research in the abstract; filling the gap in the research with presenting carry over evidence; including F-FFQ analyses in habitual diet; and the final paragraph in the Discussion: 'To our knowledge, this is the first study providing insight into the ‘niche factor’ (Hill, 2012) strategies and functional potential of B. adolescentis brought about by a 3-week intervention of a fruit juice drink with prebiotic fibres in healthy humans. Our results provide evidence of complex cross-feeding pathways and indicate areas for future research such as the importance of the gut microbiota-arginine-host axis. In addition, we add to the lacking pool of data of fibre intervention trial wash-out duration and cross-over implications and the importance of measuring habitual dietary prebiotic fibres'.
"Secondly, the number of participants for the test could be limited, and the age had a significant difference, leading to the result and conclusion could be uncertainty to some extent."
We address the participant number limitation in the following paragraph: 'The cross-over design of the study increases the power to detect changes in response to the treatment with a small number of subjects and a shorter period (Kang et al., 2022), however, the design is also responsible for the main weakness of our study considered to be the small sample size available for emotional health data processing. '
The age did not have a significant influence. This is presented in Supplemental Table 3, and we have added additional Supplemental Tables 7 - 9 demonstrating with multiple regression analyses that neither age or gender had a significant influence in our study. Please also see lines 374-375 and 414-416.
"Thirdly, the microbial community structure of these participants should be compared before and after test."
This was done, presented and discussed. See Figure 4 on page 13; Figure 5 on page 14; Table 2 on page 12; Supplemental figure 8.
"In addition, the authors cited too many references, are them necessary? If the authors have to cite these references, please put the reference in the place that you cited, not be in together." With respect, we disagree with this comment, on the contrary, one of the other reviewers complimented the manuscript on the significant update of the scientific literature. All references are in the place they are cited, where more than one reference is presented, it means they are all relevant to being cited.
"The tables and figures are too casual." With respect, it is not clear to us what the specific issue here is with our tables and figures. If the editor would like us to make specific changes to specific tables or figures, we would be happy to do so.
Yours sincerely
A. Bester

Reviewer 2 Report
General comments:
1. Interesting research.
2. Please add some background about MOJU prebiotics. E.g. how much bioactive components in it for 60 mL. Why 60 mL? This is not typical amount of ‘fruit juice drinks’ if we translate to human consumption.
3. Please check reference format throughout the text.
Title:
Ok.
Abstract:
Please add more details about MOJU drinks especially the composition.
Introduction:
Ok
Materials and Methods:
Line 124: How determined 35 sample size? By chance?
Line 128: Are they on low polyphenol diet? How make sure there was not ‘interference’ from polyphenols to the studied MOJU? Male and female included? Gender differences in gut microbiota?
Results:
Line 287: The header should be read as ‘Result and Discussion’?
Discussion:
Ok.
Conclusions:
Ok.
Author Response
Dear Reviewer, thank you very much for your valuable feedback. Please find below responses to your comments.
"Interesting research." We are indeed very appreciative of positive feedback, thank you so much.
"Please add some background about MOJU prebiotics. E.g. how much bioactive components in it for 60 mL. Why 60 mL? This is not typical amount of ‘fruit juice drinks’ if we translate to human consumption." We have updated the abstract as follows: Here, we aimed to determine the impact of the MOJU Prebiotic Shot, an apple, lemon, ginger, and raspberry fruit juice drink blend containing chicory inulin, baobab, golden kiwi, and green banana powders, on gut microbiota structure and function. Additional compositional breakdown information is presented in section 2.1.3 - providing as much as possible whilst respecting proprietary information. In addition, we have added the following sentence: The commercially available MOJU Prebiotic shot (berry) is a single serving designed as a convenient, concentrated functional ‘shot’. (line 164-5).
"Line 124: How determined 35 sample size? By chance?" Line 124 states: A total of thirty-five individuals expressed their interest in participating. Our sample size is n=20 for a cross over study. As per line 172-3: The sample size was determined based on the findings from previous studies with prebiotic fibre interventions in healthy humans (Kleesen et al., 2007; Bouhnik et al., 2007; Kolida et al., 2007; Grasten et al., 2003).
"Are they on low polyphenol diet? How make sure there was not ‘interference’ from polyphenols to the studied MOJU? Male and female included? Gender differences in gut microbiota?" Very good points you make here! We have added the following to our Discussion:
It is important to note we did not investigate the polyphenol content of the participant’s habitual diet and therefore cannot rule out a potential influence to the response. (Lines 812-3).
We have also done additional regression stats on age and gender:
Multiple linear regression demonstrated participant age and gender did not significantly influence response to the intervention for total relative abundance, p >0.05. (See Supplemental Table 7). (Lines 374-5) AND Multiple linear regression demonstrated participant age and gender did not significantly influence response to the intervention for both B. adolescentis and CAG-81SP900066785, p >0.05. (See Supplemental Tables 8 and 9). (Lines 414-6).
"Line 287: The header should be read as ‘Result and Discussion’?" This is done, see line 289.
Yours sincerely
A. Bester

Reviewer 3 Report
Cross over is a good design, but what was the power analysis to determine n? Did they have adequate power at the end of the study?
Please use the correct number of significant figures consistently. Zero should come before a decimal point.
Line 150 “(range 102-58)” not sure if there is a typo here. Or the numbers are reversed.
L300 is concerning to this reviewer. It appears they skipped over some food items.
Line 814 is concerning to this reviewer. Between inadequate power and lack of compliance for the weekly questionnaire, how do we know that this data is real.
Author Response
Dear reviewer, thank you for your valuable feedback. Please see below the response to your comments.
"Cross over is a good design, but what was the power analysis to determine n? Did they have adequate power at the end of the study? " We present in line 172-4 how we determined our sample size. The sample size was determined based on the findings from previous studies with prebiotic fibre interventions in healthy humans (Kleesen et al., 2007; Bouhnik et al., 2007; Kolida et al., 2007; Grasten et al., 2003). We believe we had adequate power at the end of the study for the gut microbiota analyses, but not for the emotional health data analyses. This we discuss in section 3.4, lines 350-358; and the Discussion section, lines 814-825.
"Please use the correct number of significant figures consistently. Zero should come before a decimal point. " Thank you for you eagle eye! We believe we have corrected all of these. There were many, and mainly concentrated across pages 9 - 12.
"Line 150 “(range 102-58)” not sure if there is a typo here. Or the numbers are reversed" Indeed, thank you for spotting this. The error has been corrected to (range 58 - 102), line 150.
"L300 is concerning to this reviewer. It appears they skipped over some food items." We did not skip over some food items. We used a validated (2011) questionnaire that asks the following:
‘Wheat, oat and barley cereals (breakfast cereals including muesli etc) – count the bowls and or cereal bars you have consumed.’ In hindsight, we should have requested a brand name. Unless the participant described the detailed product, for example oat porridge or Alpen sugar free muesli, or Kellogs strawberry cereal bar, we had no compositional information and therefore could not include these items in the analyses. By including the statement, copied below (line 302 – 304), we contribute this knowledge to make future researchers wishing to utilize the validated F-FFQ aware of the need to request more detailed information.
Composite cereal products, such as breakfast cereals and cereal bars were excluded from the calculation due to lack of detailed composite information and/or brand names. We therefore highlight the need to request this information in future studies.
"Line 814 is concerning to this reviewer. Between inadequate power and lack of compliance for the weekly questionnaire, how do we know that this data is real."
We believe we have explained above, and in the manuscript, where we have and have not adequate power. The lack of compliance for the weekly questionnaire refers to the emotional health measurements, described above. We strongly believe our data is real.
Yours sincerely
A. Bester

Reviewer 4 Report
This is a very well written but extremely detailed paper, containing many up-to-date references. Even if the findings of the research were not surprisingly un-remarkable the paper is valuable for the amount of information it contains on methods and prior research, and it seems to be an example of a thorough study.
paragraph 2 in section 2.1.1 would be helped by separating into inclusion and exclusion criteria.
Author Response
Dear reviewer, thank you for your encouraging and positive comments.
A differentiation between inclusion and exclusion criteria has been made. See lines 128 and 130.
Yours sincerely
A. Bester

Round 2
Reviewer 1 Report
Reject!!!
No.